# Crystal structure of a lipin/Pah phosphatidic acid phosphatase

Valerie I. Khayyo [1], Reece M. Hoffmann[2], Huan Wang [3], Justin A. Bell [1], John E. Burke [2], Karen Reue[3] & Michael V. Airola [1]✉

Lipin/Pah phosphatidic acid phosphatases (PAPs) generate diacylglycerol to regulate triglyceride synthesis and cellular signaling. Inactivating mutations cause rhabdomyolysis, autoinflammatory disease, and aberrant fat storage. Disease-mutations cluster within the conserved N-Lip and C-Lip regions that are separated by 500-residues in humans. To understand how the N-Lip and C-Lip combine for PAP function, we determined crystal structures of *Tetrahymena thermophila* Pah2 (*Tt* Pah2) that directly fuses the N-Lip and C-Lip. *Tt* Pah2 adopts a two-domain architecture where the N-Lip combines with part of the C-Lip to form an immunoglobulin-like domain and the remaining C-Lip forms a HAD-like catalytic domain. An N-Lip C-Lip fusion of mouse lipin-2 is catalytically active, which suggests mammalian lipins function with the same domain architecture as *Tt* Pah2. HDX-MS identifies an N-terminal amphipathic helix essential for membrane association. Disease-mutations disrupt catalysis or destabilize the protein fold. This illustrates mechanisms for lipin/Pah PAP function, membrane association, and lipin-related pathologies.

[1] Department of Biochemistry and Cell Biology, Stony Brook University, Stony Brook, NY, USA. [2] Department of Biochemistry and Microbiology, University of Victoria, Victoria, BC V8W 2Y2, Canada. [3] Department of Human Genetics, David Geffen School of Medicine at UCLA, Los Angeles, CA, USA. ✉email: michael.airola@stonybrook.edu

Triglycerides (TAG) represent the major form of long-term energy storage in eukaryotes and are synthesized by the evolutionarily conserved glycerol-3-phosphate pathway[1]. The penultimate step of TAG biosynthesis is catalyzed by magnesium-dependent phosphatidic acid phosphatases (PAPs) that hydrolyze phosphatidic acid (PA) to diacylglycerol (DAG)[2,3] (Fig. 1a). There are three lipin PAPs in humans and mice[4,5] and a single PA phosphohydrolase (Pah1) in *Saccharomyces cerevisiae*[6] (Fig. 1b).

Lipin/Pah PAPs are metabolic enzymes that regulate energy storage[4], energy mobilization[7], adipogenesis[8], phospholipid synthesis[9,10], autophagy[11], chylomicron secretion[10], and fatty acid synthesis[12]. Mutations that inactivate lipin PAP function are associated with several diseases, including rhabdomyolysis[13–18], statin-induced myopathy[11,13], the inflammatory disorder Majeed syndrome[5,19], lipodystrophy, insulin resistance, peripheral neuropathy, and neonatal fatty liver[4].

The architecture of lipin/Pah PAPs varies, but all PAPs conserve two regions referred to as the N-Lip and C-Lip[2,3], originally named for their location on opposite termini of human and mouse lipins[4] (Fig. 1b). The C-Lip is predicted to contain the catalytic domain from the haloalkanoic acid dehalogenase (HAD) superfamily[20] and harbors a catalytic DxDxT motif[6,21]. The N-Lip is required for PAP activity[21–23], but its function is unclear. Functional prediction for the N-Lip has been hindered by a lack of sequence homology with domains of known function and the N-Lip being uniquely found in PAPs. Separating the N-Lip and C-Lip is an extended linker that typically contains 250–500 residues[2,3]. The linker sequence is highly variable, but retains a conserved function to regulate lipin/Pah PAP localization and activity[2,22,24–28].

Despite nearly 20 years of research, the role of the N-Lip and how it combines with the C-Lip to form a productive PAP enzyme has been unclear. Here we determined the first crystal structures of a lipin/Pah PAP from *Tetrahymena thermophila* that revealed a two-domain architecture with a HAD-like catalytic domain and an unexpected split immunoglobulin-like domain formed by the N-Lip and C-Lip. Complementary activity assays, biophysical, and cellular experiments explain how the interplay between these domains, with an N-terminal amphipathic helix, facilitates lipin membrane association and catalysis. This work also explains how disease-associated mutations inactivate this important metabolic enzyme.

## Results

**Minimal lipin/Pah PAPs.** To establish a minimal functional core of lipin/Pah PAPs, we used domain-enhanced BLAST searches and identified lipin/Pah PAPs in some ciliate and plant species with the essential N-Lip and C-Lip regions directly fused (Supplementary Fig. 1a). These minimal lipin/Pah PAPs lack the extended linkers in mammalian lipins and *S. cerevisiae* Pah1 that regulate their cellular localization and activity (Fig. 1b). Six minimal lipin/Pah PAPs were cloned from cDNA or gene synthesized (Supplementary Fig. 1a). Three were tested for complementation of the growth defects of *Sc pah1Δ* cells on agar plates[21]. All three complemented the ability of *Sc pah1Δ* cells to utilize glycerol as the sole carbon source (Supplementary Fig. 1b), but none complemented the heat stress phenotype. During the course of our study, a minimal lipin/Pah from the ciliate *T. thermophila* (*Tt*) (known as *Tt* Pah2) was reported to also complement the glycerol phenotype, but not the heat

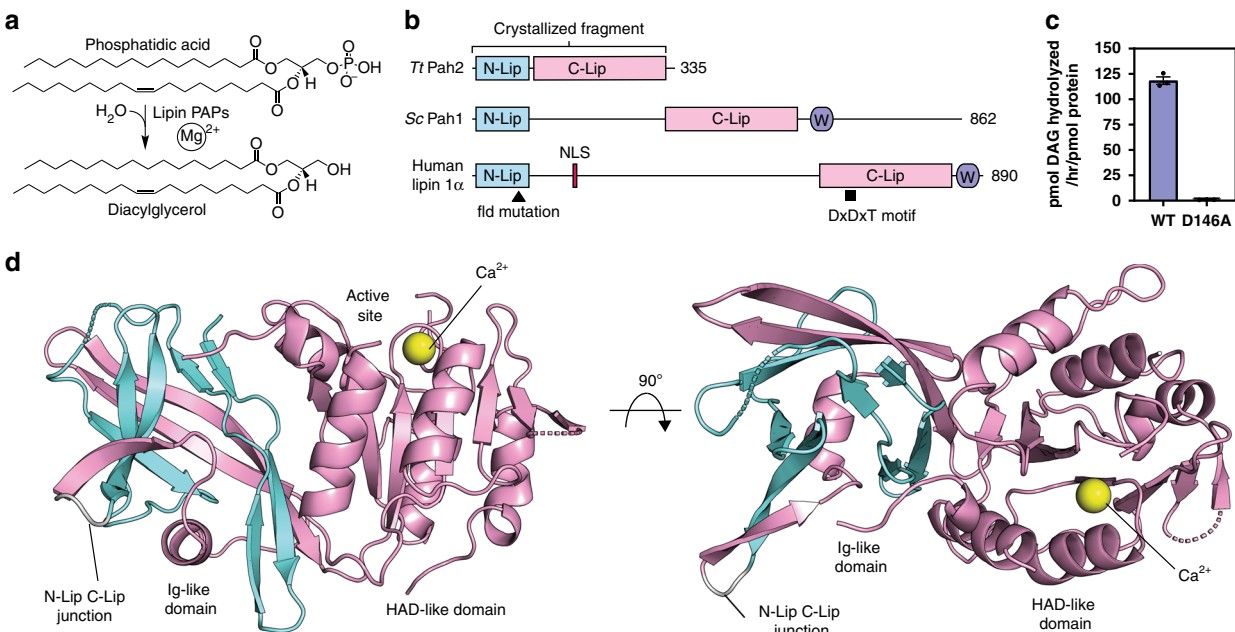

**Fig. 1 Structure of *Tt* Pah2 reveals the N-Lip co-folds with the C-Lip to form a split immunoglobulin-like domain. a** Phosphatidic acid hydrolysis by magnesium-dependent lipin PAPs produces diacylglycerol. **b** Domain architecture of PAPs drawn to scale. *Tt* Pah2 directly fuses the N-Lip and C-Lip regions that are located on opposite termini of human lipins and separated by 250 residues in *Sc* Pah1. The positions of the nuclear localization signal (NLS), conserved Trp-motif (purple W), catalytic DxDxT motif, and fatty liver dystrophy (fld[2J]) mutation are indicated. **c** Wild-type (WT) *Tt* Pah2 is catalytically active and the D146A mutant of the DxDxT motif eliminates activity. Error bars represent standard deviation (*n* = 3) of experiments performed in technical duplicates. **d** Overall structure of *Tt* Pah2. *Tt* Pah2 contains two domains: an immunoglobulin-like (Ig-like) domain and a HAD-like catalytic domain. The Ig-like domain is formed by the N-Lip (cyan) and C-Lip (pink) regions connected by a short linker (gray loop) that would be replaced by the extended 500-residue linker in human lipins and a 250-residue linker in *Sc* Pah1. A calcium ion (Ca[2+], yellow sphere) is bound in the active site of the HAD-like catalytic domain. Top view (right) of the enzyme shows the N-Lip co-folding with the C-Lip to form the Ig-like domain. Source data are provided as a Source Data file.

stress phenotype of *Sc pah1Δ* cells[29,30]. We subsequently crystallized *Tt* Pah2.

*Tt* Pah2 shares 36 and 42% sequence identity with the N-Lip and C-Lip regions of human lipins (Supplementary Fig. 2), and is an active PAP in vitro (Fig. 1c). In HEK293 cells, *Tt* Pah2 localized exclusively to the endoplasmic reticulum (ER) independent of fatty acid treatment, which is known to cause translocation of mammalian lipins to the ER[31,32] (Supplementary Fig. 3). The crystallized *Tt* Pah2 construct (residues 1–321), which lacks 14 C-terminal residues, also localized to the ER (Supplementary Fig. 3). In contrast, mouse lipin 1 localized to the nucleus under basal conditions, and partially translocated to the ER only after fatty acid treatment (Supplementary Fig. 3). This supports the hypothesis that the localization of *Tt* Pah2 to the ER may be constitutive, which contrasts with mammalian lipins and may relate to the presence of the linker sequence.

**Overall structure**. *Tt* Pah2 was produced in *Escherichia coli* and crystal structures were refined to 3.0 Å resolution in the presence of either calcium or magnesium (Table 1). Phases were determined using single-wavelength anomalous diffraction from selenomethionine-enriched protein crystallized with calcium. The structure revealed a two-domain architecture with an immunoglobulin-like domain (Ig-like domain, residues 20–139) and a catalytic domain (residues 140–321) belonging to the haloalkanoic acid dehalogenase (HAD) superfamily (Fig. 1d). Electron density was not observed for the first 19 N-terminal residues, as well as several small loops in the catalytic domain. Notably, the N-Lip did not form its own domain, but rather cofolded with the first 45 residues of the C-Lip to form the Ig-like domain, with the remainder of the C-Lip forming the HAD-like catalytic domain (Fig. 1d).

**Catalytic domain**. The *Tt* Pah2 HAD-like catalytic domain is characterized by a Rossmann-like fold with two peripheral antiparallel β-strands (β6 and β7) that create a central 7 stranded mixed β-sheet, flanked by 5 α-helices (Figs. 1d, 2a). The HAD superfamily is

represented by >30,000 different members that act on different substrates. The HAD superfamily commonly has large domains (~100–200 residues) inserted between topological elements referred to as "caps", which are classified into C1 and C2-type caps. C1 cap domains are inserted after the β1 strand, while C2 cap domains are inserted after the β3 strand. In both cases, these cap domains reside above the HAD core and help form a structural pocket above the active site to generate substrate specificity[20,33].

Notably, the HAD-like catalytic domain of *Tt* Pah2 is "capless" with an active site completely exposed to solvent (Figs. 1d, 2a) and available to interact with the membrane during interfacial catalysis. The shallow and solvent exposed active site of *Tt* Pah2 implies that the specificity of lipin/Pah PAPs for PA[5] is most likely driven by interaction with the polar glycerol backbone of PA and not through interaction with the membrane-embedded fatty acid tails.

There are short peptides in *Tt* Pah2 inserted in the C1 and C2 cap positions that could participate in PA recognition. The C1 peptide could not be modeled due to a lack of electron density, but immediately follows the catalytic DxDxT motif and would lie in close proximity to the active site (Fig. 2a). The C2 peptide forms a helix away from the active site, near the membrane interface, but is only visible in the magnesium structure, likely due to its involvement in a crystal contact (Fig. 2a). While unlikely, the C2 peptide may be able to adopt a different conformation to participate in PA recognition.

**Immunoglobulin-like domain**. Adjacent to the catalytic domain is an atypical Ig-like domain that packs against and stabilizes the catalytic domain to create an expanded flat surface at the membrane interface (Fig. 1d). Canonical Ig domains have seven β-strands, labeled A–G, which form a two-layered β-sandwich (Fig. 2b). The Ig-like domain of *Tt* Pah2 differs significantly from canonical Ig domains. The most prominent difference occurs at the βE strand, which splays outward away from the β-sandwich core and forms a β-hairpin with a βE′ strand (Fig. 2b). A small helical insert immediately following the βE′ strand connects back to the canonical βF and βG strands to complete the β-sandwich.

Notably, the transition between the N-Lip and C-Lip occurs directly at the two-residue turn in the β-hairpin formed by the βE and βE′ strands (Fig. 2b). In *Sc* Pah1 and human lipins, this two-residue turn is replaced by the extended 250–500 residue linker that separates the N-Lip and C-Lip regions (Fig. S2). This implies that in human lipins and *Sc* Pah1, the Ig-like domain is a "split" domain that must recombine to form a functional PAP enzyme.

**Catalytic mechanism**. Key active site residues to hydrolyze PA to DAG are clustered at the putative membrane interface in the typical topological positions of HAD members (Fig. 3a). In the structure determined in the presence of magnesium, a Mg²⁺ ion was modeled in the typical position for HAD phosphatases with correct geometry and distances expected for Mg²⁺ coordination. This Mg²⁺ ion is coordinated by the sidechain of the first Asp residue of the DxDxT motif (Asp146), as well as by the sidechain of Asn268 and the mainchain carbonyl oxygen of Asp148. As expected in the absence of the substrate PA, the sidechain of Asp148 (the second Asp residue of the DxDxT motif) is directed away from the Mg²⁺ ion and forms a salt bridge with Arg193. Other residues predicted to stabilize the trigonal bipyramidal transition state[20,34] are present in the expected positions, with Lys244 directed towards the Mg²⁺ ion and Ser191 adjacent (Fig. 3a).

The structure suggests that lipin/Pah PAPs utilize a similar catalytic mechanism as proposed for other HAD phosphatases[20,34] with Asp146 serving as a nucleophile to attack PA and form the trigonal bipyramidal transition state, the surrounding

**Table 1 Data collection and refinement statistics.**

| | Se-Met Tt Pah2 Calcium | Tt Pah2 Calcium | Tt Pah2 Magnesium |
|---|---|---|---|
| **Data collection** | | | |
| Space group | C 1 2 1 | C 1 2 1 | C 1 2 1 |
| Cell dimensions | | | |
| *a, b, c* (Å) | 107.53, 135.42, 91.61 | 108.02, 135.13, 90.97 | 165.41, 57.64, 84.19 |
| *α, β, γ* (°) | 90, 116.3, 90 | 90, 116.19, 90 | 90, 114.9, 90 |
| Wavelength | 0.9794 | 0.9793 | 0.9184 |
| Resolution (Å) | 43.38–3.25 (3.34–3.25)* | 52.05–3.0 (3.107–3.0) | 44.89–3.0 (3.107–3.0) |
| $R_{merge}$ | 0.174 (1.47) | 0.0888 (0.4502) | 0.1996 (0.5586) |
| *I/σI* | 9.21 (1.68) | 13.27 (3.98) | 5.78 (2.60) |
| Completeness (%) | 100 (100) | 99.95 (100.00) | 99.72 (99.79) |
| Redundancy | 13.9 (13.6) | 6.9 (7.1) | 6.8 (6.4) |
| **Refinement** | | | |
| Resolution (Å) | | 3.0 | 3.0 |
| No. reflections | | 162667 | 100293 |
| $R_{work}/R_{free}$ | | 0.2317/0.2656 | 0.2419/0.2828 |
| No. atoms | | 8653 | 4397 |
| Protein | | 8590 | 4366 |
| Ligand/ion | | 5 | 2 |
| Water | | 58 | 29 |
| *B*-factors | | | |
| Protein | | 85.98 | 70.38 |
| Ligand/ion | | 101.07 | 57.66 |
| Water | | 68.00 | 42.34 |
| R.m.s. deviations | | | |
| Bond lengths (Å) | | 0.006 | 0.005 |
| Bond angles (°) | | 0.94 | 1.15 |

*Values in parentheses are for highest-resolution shell.

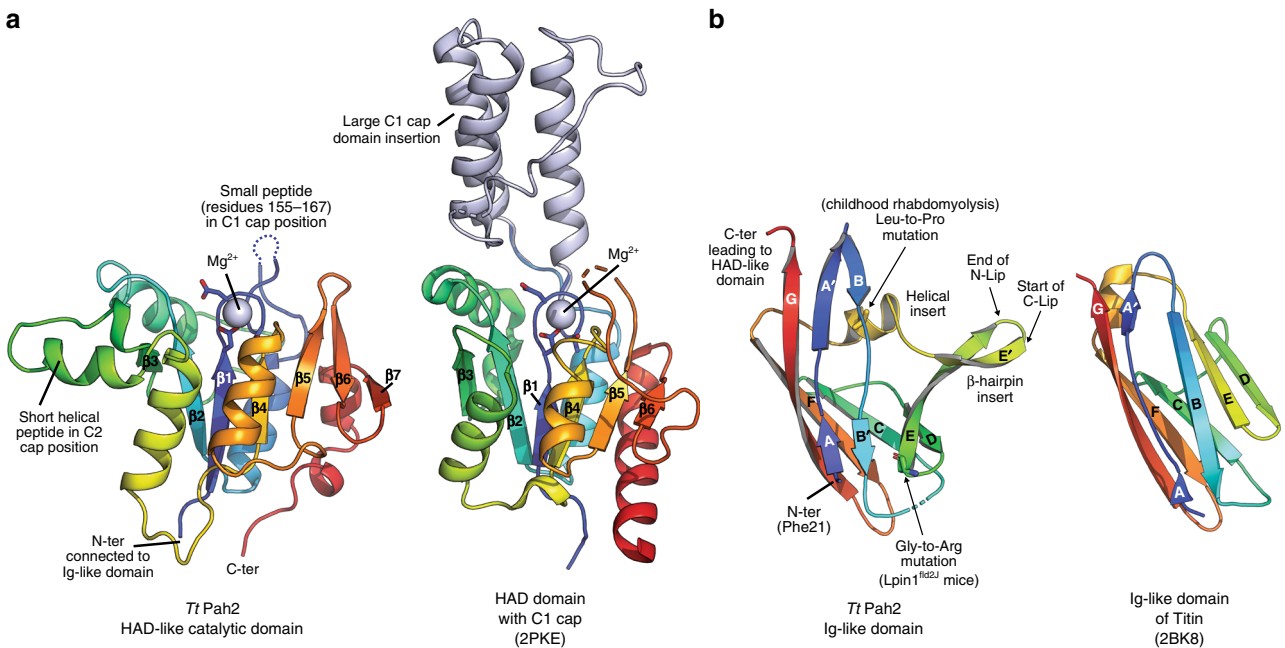

**Fig. 2 Structural comparison of HAD and Ig domains with *Tt* Pah2. a** Structural comparison of the *Tt* Pah2 HAD-like domain with a C1-type capped member of the HAD superfamily. The *Tt* Pah2 HAD-like domain is capless with two short peptides in lieu of large cap domain insertions to allow close association with membranes. Magnesium ions are shown as light blue spheres. **b** The Ig-like domain of *Tt* Pah2 forms an atypical Ig-like β-sandwich fold with the βE-strand at the end of the N-Lip splaying outward and forming a β-hairpin with the βE′ strand at the start of the C-Lip. A helical insert follows, which contains a conserved Leu residue that is mutated in patients with childhood rhabdomyolysis packing into a hydrophobic core. The conserved glycine residue mutated in Lpin1^{fld2J} mice is located at the beginning of βE-strand. The more typical Ig-like fold of the muscle protein Titin is shown for comparison with the β-strands labeled A–G.

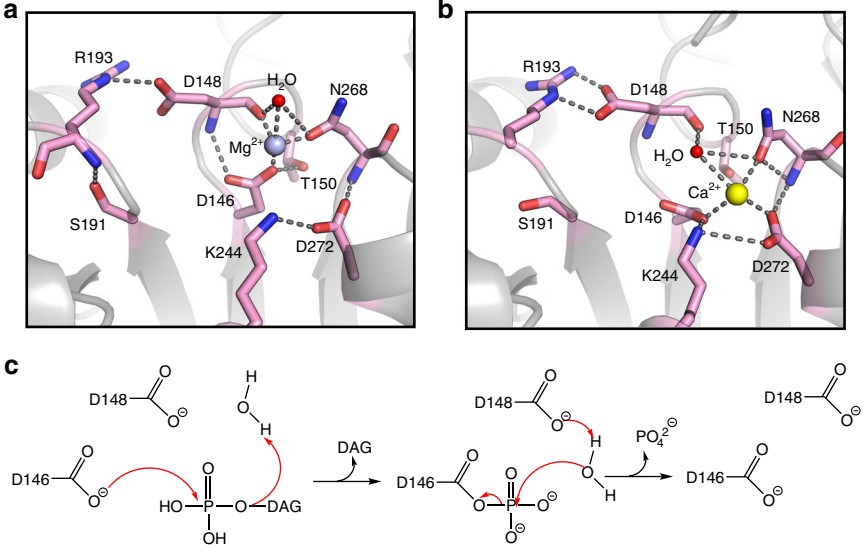

**Fig. 3 Active site and catalytic mechanism. a** Detailed view of the *Tt* Pah2 active site with magnesium bound (light blue sphere). Polar contacts are indicated by gray dashes. **b** Detailed view of the structural rearrangement of key residue interactions in the active site with calcium bound (yellow sphere). **c** Proposed schematic of lipin/Pah PAPs catalytic mechanism. Catalysis proceeds through a trigonal bipyramidal intermediate.

residues stabilizing the transition state through electrostatic interactions, and Asp148 activating a water molecule for nucleophilic attack to liberate DAG and inorganic phosphate (Fig. 3c).

In the structure determined in the presence of calcium, a $Ca^{2+}$ ion was modeled in the active site in an altered position in comparison to the $Mg^{2+}$ ion, with Asp272 coordinating the $Ca^{2+}$ ion, along with the sidechains of Asp146 and Asn268 (Fig. 3b).

The altered location of the $Ca^{2+}$ ion and subsequent rearrangement of key active site residues could prevent proper binding and hydrolysis of PA, which is consistent with the magnesium dependence of lipin/Pah PAPs.

**Membrane association.** In vitro mammalian lipins and *Sc* Pah1 interact strongly with liposomes containing the anionic lipid

substrate, PA[24,35–37]. *Tt* Pah2 also preferentially interacted with PA-containing liposomes, in comparison with neutral or anionic PS-enriched liposomes (Fig. 4a, b).

To identify the membrane-binding regions in *Tt* Pah2 and the conformational changes that occur during membrane interaction, we employed hydrogen-deuterium exchange mass spectrometry (HDX-MS) in the absence and presence of PA-containing liposomes. The rate of deuterium exchange for the amide hydrogens is an excellent readout of secondary structure dynamics and can be used to identify regions with altered conformations between states[38]. This technique has been particularly useful in identifying protein-membrane interfaces, and lipid-induced conformational changes[39,40].

To avoid hydrolysis of the PA substrate, HDX-MS experiments were conducted with the D146A mutant of *Tt* Pah2, which was catalytically inactive (Fig. 1c) even at the µM protein concentrations required for HDX-MS. HDX-MS experiments were carried out at five timepoints of exchange (3, 30, 300, 3000, 10,000 s). Critical to the analysis of an HDX-MS experiment is the generation of a peptide map that allows for localization of deuterium incorporation. A total of 193 peptides spanning 98.8% of the primary sequence were identified and quantified. All deuterium incorporation data for all timepoints can be found in the source data (Supplementary file).

HDX-MS revealed multiple regions that were protected from amide exchange in the presence of liposomes (N-terminus 1–17,

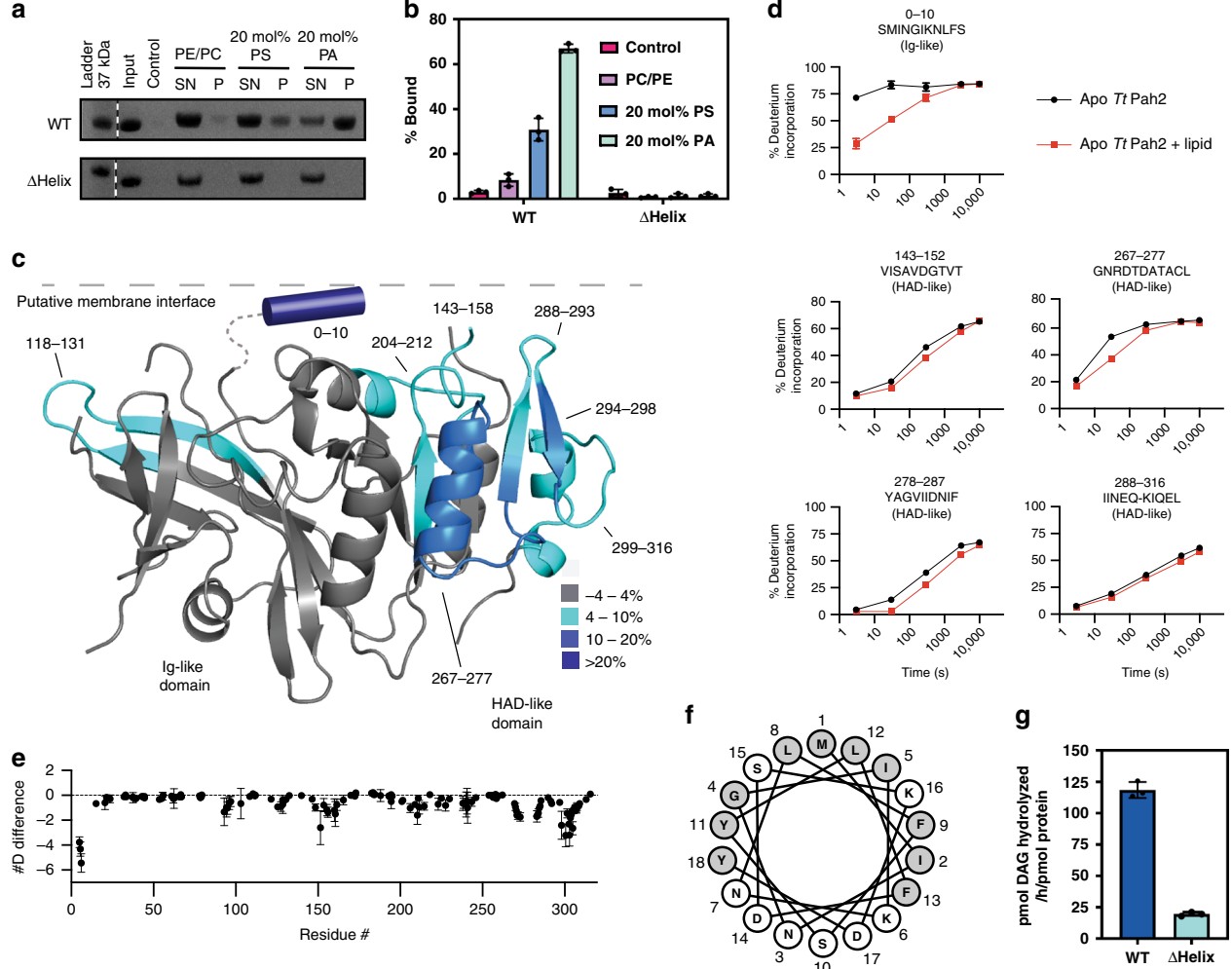

**Fig. 4 HDX-MS identifies a putative amphipathic helix for Pah2 membrane association. a** SDS-PAGE analysis of liposome sedimentation assay reveals wild-type *Tt* Pah2 preferentially associates with liposomes containing 20 mol% PA. Deletion of the putative amphipathic helix in ΔHelix *Tt* Pah2 results in loss of membrane association. SN supernatant, P pellet, PE phosphatidylethanolamine, PC phosphatidylcholine, PS phosphatidylserine, PA phosphatidic acid. **b** Quantification of liposome association for wild-type and ΔHelix *Tt* Pah2. Error bars represent standard deviation (n = 3). **c** Regions of *Tt* Pah2 that showed significant decreases in exchange (defined as >4%, >0.3 Da, and a Student's t test p < 0.01) in the presence of 20% PA liposomes are colored in blue according to the legend. Peptides around the active site and a putative N-terminal amphipathic helix (blue cylinder) show the most protection from deuterium exchange. **d** % deuterium incorporation of selected peptides at various timepoints (3, 30, 300, 3000, and 10,000 s). The strong protection of the peptide comprising residues 0–10 is suggestive of both a decrease of solvent exposure (due to membrane binding) and the formation of new secondary structure in the presence of membranes. The error bars represent standard deviation (n = 3), most are smaller than the size of the point. **e** The sum of the # of deuterons protected from exchange in the presence of membranes across all timepoints is shown. Each point represents a single peptide, with them being graphed on the x-axis according to its central residue. Error bars represent standard deviation (n = 3). **f** Helical wheel diagram of residues 1–18 in *Tt* Pah2 that were disordered in the crystal structure. Hydrophobic residues colored gray, polar residues colored white. **g** Deletion of the amphipathic helix in ΔHelix *Tt* Pah2 disrupts PAP activity. Error bars represent standard deviation (n = 3) of experiments performed in technical duplicates. Source data are provided as a Source Data file.

118–131, 143–158, 204–212, and 267–316) (Fig. 4c, d). As expected, several peptides around the active site were protected, consistent with their interaction with the membrane for productive catalysis (Fig. 4c, d). One peptide in the Ig-like domain was protected (Fig. 4c). This suggests the Ig-like domain contributes at least partially to membrane binding.

The most striking and largest decrease in exchange induced by membranes was within the first nine residues of the N-Lip region, which were disordered in the crystal structure (Fig. 4c–e). This substantial protection (Fig. 4d) most likely indicates formation of new secondary structure, such as an amphipathic helix. This is consistent with previous reports of an N-terminal amphipathic helix in *Sc* Pah1[35] and also suggested by a helical wheel diagram of *Tt* Pah2 (Fig. 4f).

As amphipathic helices commonly associate with membranes[41], we speculated this region of *Tt* Pah2 would be critical for membrane association and PAP activity. Accordingly, deletion of the putative amphipathic helix completely eliminated membrane association with liposomes (Fig. 4a, b) and reduced the catalytic activity of *Tt* Pah2 by ~80% (Fig. 4g). Taken together, this suggests an N-terminal amphipathic helix is a conserved feature of lipin/Pah PAPs that drives association to PA-enriched membranes.

**Structural mapping of disease mutations**. *Tt* Pah2 conserves the native residues that are mutated in several lipin-related pathologies (Fig. 5a, Supplementary Fig. 2). Four mutations map to the HAD-like domain. Three mutations directly disrupt catalysis: both R193H (R725H in human lipin 1[17,18]) and S191L (S734L in human lipin 2[19]) affect stages of the catalytic cycle, and G267R (G799R in human lipin 1[17]) lies adjacent to Asn268, which

coordinates the magnesium ion (Fig. 5b, c). Y306 plays a key structural role, packing into the hydrophobic core of the HAD-like domain (Fig. 5b). Consistently, the point mutant Y306N (Y873N in mouse lipin 1[42]) affects both the PAP activity (Fig. 5c) and overall stability of *Tt* Pah2 (Fig. 5d).

L103P (L635P in human lipin 1[16]) is the only C-Lip mutation that maps to the Ig-like domain (Fig. 5a). L103 is located in the helical insert of the Ig-like domain and makes stabilizing hydrophobic interactions with the N-Lip portion of the Ig-like domain (Fig. 5b). Mutation to proline would both break the helix and eliminate the hydrophobic interactions made by the Leu sidechain. In *Tt* Pah2, this mutation reduces PAP activity by 90% (Fig. 5c) and decreases the melting temperature by 5° (Fig. 5d).

G79R is analogous to the G84R missense mutation originally described in *Lpin1*[fld2J] mice[4], and is the only mutation in the N-Lip. Surprisingly, G79R had no affect on the stability, PAP activity, or membrane association of *Tt* Pah2 (Fig. 5c–f). Structurally, this glycine completes a β-turn between the βD and βE strands (Fig. 5b), with Phi and Psi angles in the expanded allowed region for glycine residues. Potential reasons for the retention of full activity for G79R are discussed in more detail below.

**Catalytic unit of mammalian lipins**. To support the notion that mammalian lipin PAPs function with the same domain architecture observed in *Tt* Pah2, we generated analogous constructs of mouse lipin 1 and mouse lipin 2 that directly fused the N-Lip and C-Lip regions (Fig. 6a). We focused on characterizing the mouse lipin 2 construct because it was more easily purified from *E. coli* (Supplementary Fig. 6). The mouse lipin 2 N-Lip C-Lip fusion

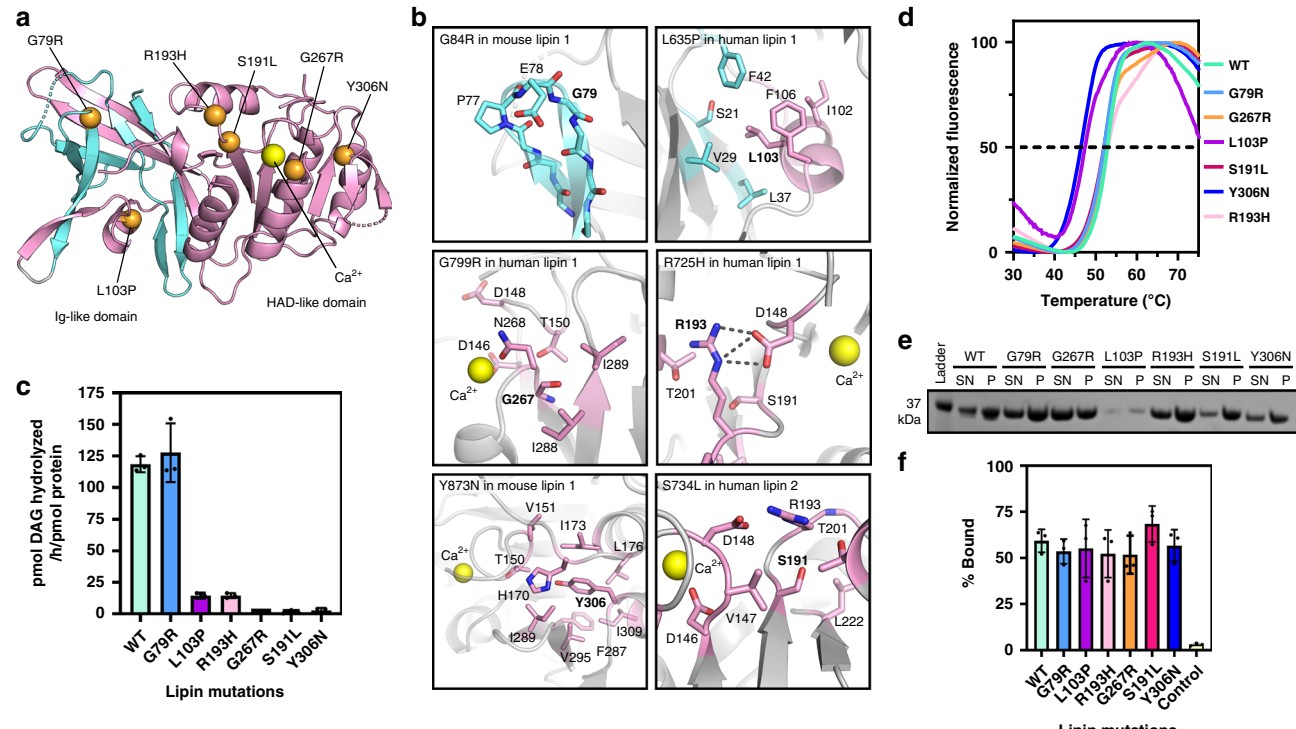

**Fig. 5 Structural mapping of disease mutations. a** Mapping of disease-associated mutations (orange spheres) in human and mouse lipins on the *Tt* Pah2 structure. A yellow sphere indicates the calcium ion. **b** Detailed view of the structural interactions of native residues in *Tt* Pah2 that are mutated in lipin-related pathologies. **c** Affects of disease mutations on the PAP activity of *Tt* Pah2. Error bars represent standard deviation (n = 3) of experiments performed in technical duplicates. **d** Effects of disease mutations on the thermal stability of *Tt* Pah2 protein assessed by differential scanning fluorimetry. Data are representative of two independent experiments. **e** SDS-PAGE analysis of liposome sedimentation assay with wild-type *Tt* Pah2 and disease-associated mutants. **f** Quantification of liposome association for wild-type *Tt* Pah2 and disease mutants expressed as % bound. Data are the means and SDs of three experiments. Source data are provided as a Source Data file.

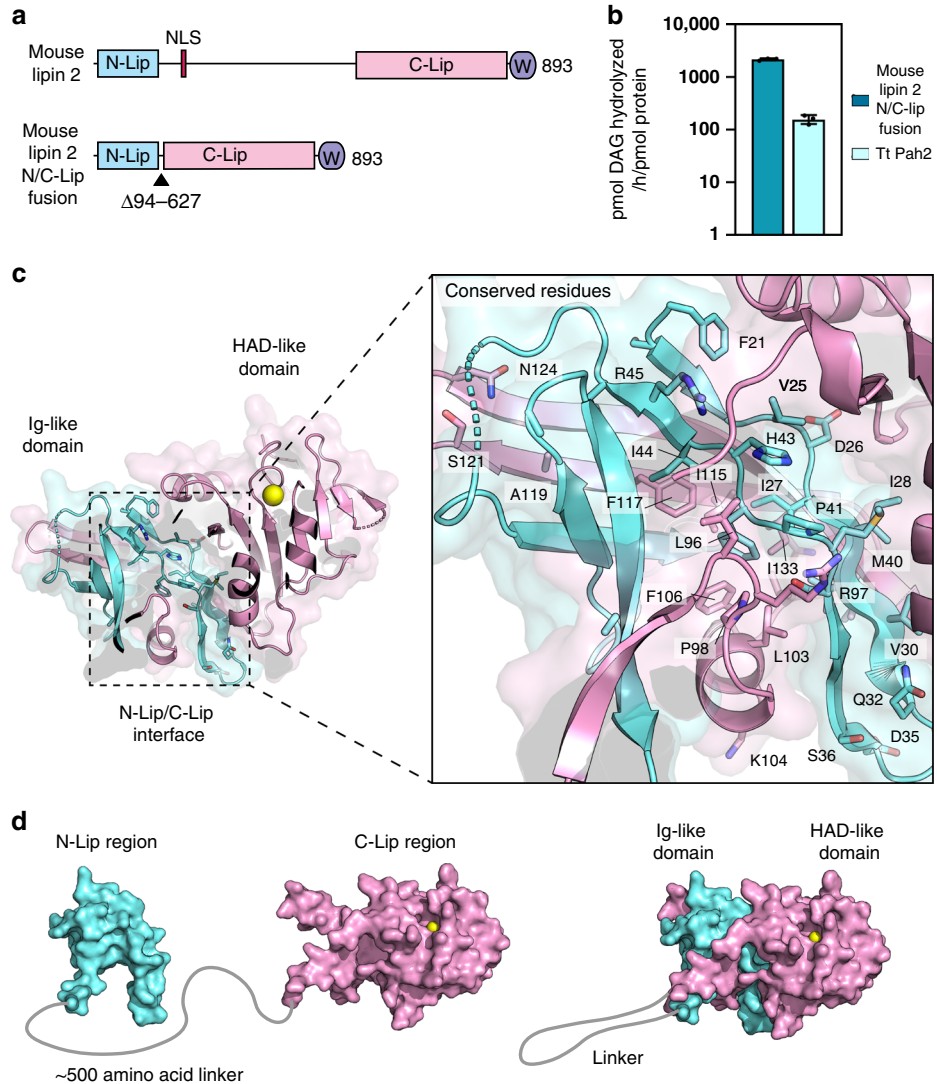

**Fig. 6 The N-Lip and C-Lip regions are the functional catalytic unit of lipin/Pah PAPs. a** Domain architecture of full-length mouse lipin 2 and an N-Lip C-Lip fusion (residues 94−627 deleted) with an analogous architecture to *Tt* Pah2. The positions of the nuclear localization signal (NLS, red) and conserved Trp-motif (W, purple) are shown. **b** Mouse lipin 2N/C-Lip fusion is catalytically active and one order of magnitude greater than *Tt* Pah2. Error bars represent standard deviation (*n* = 3) of experiments performed in technical duplicates. **c** Detailed view of homologous residues at the N-Lip and C-Lip interface. Highlighted residues are shown as sticks and share sequence identify or sequence similarity between *Tt* Pah2, *Sc* Pah1, and lipins 1, 2, and 3. **d** Proposed structure of full-length lipins. The lipin Ig-like domain is a "split" domain divided by a ~500 amino acid linker that separates the N-Lip and C-Lip. The two regions must recombine to form a functional PAP enzyme. Source data are provided as a Source Data file.

protein retained PAP activity in vitro and was an order of magnitude more active than *Tt* Pah2 (Fig. 6b). Notably, a similar construct of *Sc* Pah1 that fuses the N-Lip and C-Lip regions has been shown to retain PAP activity in vitro, as well as maintain cellular function if a Trp-containing motif that is C-terminal to the C-Lip is retained[43]. Together, this suggests the functional catalytic unit of lipin/Pah PAPs is represented by the domain architecture observed in *Tt* Pah2 with the N-Lip and C-Lip domains co-folding to form the Ig-like and HAD-like domains. In support of this, the major structural interfaces between the N-Lip and C-Lip regions are largely conserved between *Tt* Pah2, mammalian lipins, and *Sc* Pah1 (Fig. 6c) as observed by near complete sequence conservation of the core regions of the Ig-like fold, such as the βA, βA′, βB, and βB′ strands of the N-Lip (Supplementary Fig. 2b).

## Discussion

Prior to this study, the molecular mechanism of the N-Lip in lipin/Pah PAP function had remained an enigma for decades.

Here we establish that the N-Lip co-folds with the C-Lip to form a unique and rare cytoplasmic Ig-like domain. This establishes a clear role for the N-Lip, which resides on the opposite end from the enzymatic domain in mammalian lipins, in forming a catalytically active PAP enzyme. It further establishes that the N-Lip and C-Lip are sufficient for PAP function in vitro[43] and suggests the elaborated and varied architecture observed in canonical lipin/Pah PAPs has evolved to allow controlled regulation of PAP localization and activity in cells.

The structure presented here almost certainly represents the active state of lipin/Pah PAPs with the Ig-like domain packing against and stabilizing the HAD-like domain. What is not clear is whether the N-Lip and C-Lip are constitutively associated in mammalian lipins and *Sc* Pah1 or if their interaction is regulated. It is known that posttranslational modifications within the 500-residue linker that separates the N-Lip and C-Lip in humans and yeast can regulate both lipin/Pah PAP activity and subcellular localization[22,24–28]. It is therefore tempting to speculate that

modifications within the linker may prevent formation of the Ig-like domain (Fig. 6d), and therefore inactivate PAP activity or prevent membrane association. Further experimentation is required to address this hypothesis.

An N-terminal amphipathic helix in *Sc* Pah1 was previously proposed to regulate membrane binding[35]. Here we provide direct experimental evidence to support this structural feature with HDX-MS revealing that membranes induce significant deuterium protection of the N-terminus, which is consistent with formation of new secondary structure. This inducible formation of a membrane-binding amphipathic helix is not common, but is notably similar to the membrane sensing mechanism of the lipid biosynthetic enzyme CTP:phosphocholine cytidylyltransferase (CCT)[44]. While the hydrophobic face of the amphipathic helix of *Tt* Pah2 clearly interacts with the membrane, it is not clear if the hydrophilic face of this helix packs against the protein or remains exposed to solvent.

Lipin/Pah PAPs are peripheral membrane enzymes that are recruited primarily to ER membranes to regulate phospholipid and triglyceride levels[1]. This work further establishes a conserved lipid binding specificity of PAPs with *Tt* Pah2, human lipins, and *Sc* Pah1 all demonstrating a preference for PA-enriched membranes[24,35]. This is intriguing given that PA concentration in ER membranes is quite low[45], but may reflect a regulatory feature to recruit lipins to the ER when PA concentrations are elevated beyond a certain threshold.

To our knowledge, *Tt* Pah2 represents the first structurally defined lipid metabolizing member of the HAD superfamily. The capless nature of the *Tt* Pah2 HAD domain is reminiscent of other capless HAD enzymes whose substrates are large macromolecules[20]. Similarly, lipin/Pah PAPs require a solvent exposed active site to associate with the membrane surface during interfacial catalysis. Thus, we predict other lipid modifying HAD members will also be capless to facilitate access to their membrane-bound substrates.

Mammalian lipins are known to oligomerize into large assemblies with the N-Lip and C-Lip mediating oligomerization through N-Lip-to-N-Lip and C-Lip-to-C-Lip interactions[46]. We did not observe any direct evidence of this in *Tt* Pah2, where the N-Lip and C-Lip regions are directly fused. It is plausible that in human lipins, the N-Lip from one lipin molecule could interact in *trans* with the C-Lip of a second lipin molecule to form the Ig-like domain and induce oligomerization. However, this oligomerization has not been observed in *Sc* Pah1. Therefore we expect mammalian lipins may contain an additional dimerization domain that enables formation of larger oligomers.

The structure of *Tt* Pah2 provides a much deeper understanding for the clinically relevant mutations in lipin-related pathologies. Most intriguing is the C-Lip L635P missense mutation in human Lipin1 that results in childhood rhabdomyolysis[16]. Here we establish that this mutation would disrupt formation of the Ig-like domain, providing a mechanism to explain lipin dysfunction in these patients. Other mutations directly affect the catalytic machinery or destabilize the HAD-like fold. The retention of full activity of the analogous mutation in Lpin1[fld2J] mice was unexpected, as HA-tagged mouse lipin 1[fld2J] with the G84R mutation retains only 20% activity[22]. This is most likely due to local differences in structure between *Tt* Pah2 and mouse lipin 1 near the βE strand that allows accommodation of this mutation in *Tt* Pah2 without affecting function.

## Methods

**Tt Pah2 plasmids**. Residues 1–321 of the gene encoding *T. thermophila* Pah2 (*Tt* Pah2, accession number: XP_001020467.1) were codon optimized for expression in *E. coli*, gene synthesized (BioBasic, Canada), and inserted into pET28 and ppSUMO plasmids, which contain an N-terminal His-tag or N-terminal His-tag

followed by the SUMO gene. Point mutations and truncations of *Tt* Pah2 were made in the ppSUMO vector by standard PCR, the QuikChange method, or overlap extension. For mammalian expression, *Tt* Pah2 constructs were subcloned into pcDNA3.1 (Invitrogen) with a C-terminal V5-His-tag. The 14 additional C-terminal residues in wild-type *Tt* Pah2 (residues 1–335) were added back to the pcDNA construct by PCR. All constructs were verified by direct sequencing.

**Minimal lipin/Pah plasmids**. Three minimal lipins: *S. tuberosum* lipin (XP_006354896), *N. tabacum* lipin 1a (XP_016447116), and *N. tabacum* lipin 1b (XP_016447115) were PCR amplified from cDNA and cloned into pET28 and ppSUMO plasmids for *E. coli* expression (Agilent Technologies, Cat. No. 230280). These three minimal lipins were subcloned into the yeast expression vector YEp351 that contained the endogenous 5′ and 3′ untranslated ends of the Pah1 gene by removing the Pah1 gene from the vector pGH312. Two additional minimal lipins, *I. multifiliis* lipin (XP_004037123, residues 1–321) and *Capsicum annuum* lipin (XP_016568757, residues 1–321), were codon optimized for expression in *E. coli*, gene synthesized (BioBasic, Canada), and inserted into pET28 and ppSUMO plasmids.

**Mouse lipin 2 N-Lip C-Lip fusion plasmid**. A construct of mouse lipin 2 comprising residues 1–893 Δ96-627 was cloned using the overlap extension method and cloned into ppSUMO using BamHI and NotI restriction sites.

**Yeast spot assays**. *Saccharomyces cerevisiae* GHY57 cells[6] (Matα ade2-1 can1-100 his3-11,15 leu2-3,112 trp1-1 ura3-1 pah1 Δ::URA3) (a gift from Gil-Soo Han and George Carman) were transformed with lipin/YEp351 plasmids by the lithium acetate method and selected on agar plates with Leu- synthetic dropout (SD) media supplemented with 2% glucose at 30 °C. For spot assays, *S. cerevisiae* cells were grown to saturation in SD Leu- media with 2% glucose, washed with sterile water, and diluted to an optical density at 600 nm of 1. Serial dilutions (1:10) of the cells were spotted (5 μL) onto SD Leu- plates with 2% glucose or 2% glycerol, and growth was scored after 3 days of incubation at 30 °C.

**Cell culture and transfection**. Human embryonic kidney 293 cells (HEK293, American Type Culture Collection #CRL-1573) were maintained in minimum essential medium as specified by the supplier. All experiments were performed using cells with 2–6 passages. To investigate the subcellular location of *Tt* Pah2, cells were transfected with expression constructs for *Tt* Pah2 1–321, *Tt* Pah2 1–335, or full-length mouse lipin-1 with C-terminal V5 epitopes using BioT reagent according to the manufacturer's protocol (Bioland Scientific LLC). Thirty-six hours after transfection, cells were treated with either 400 μM oleic acid or BSA (as control) for 24 h before immunostaining.

**Immunostaining and confocal immunofluorescence imaging**. Cells were fixed in 4% paraformaldehyde for 10 min, permeabilized with 0.2% Triton X-100 for 10 min, and treated with 10% serum for 30 min at room temperature. Primary antibodies including mouse monoclonal V5 (1:500 dilution, Invitrogen, Cat. 960-25) and Calnexin (1:1000 dilution, Abcam, Cat. ab22595) were applied and incubated at 4 °C overnight, followed by incubation with relevant secondary antibodies (1:500 dilution, Alexa 647-labeled donkey-anti-mouse, Cat. A31571; Alexa 488-labeled donkey-anti-rabbit, Cat. A21206, Invitrogen) for 1 h in a dark chamber at room temperature. Coverslips were mounted using ProLong Gold Antifade Mountant with 4′,6-diamidino-2-phenylindole (Invitrogen, Cat. P36931). Images were generated with a Leica confocal SP8-STED/FLIM/FCS laser-scanning microscope equipped with an argon-krypton laser at a magnification of ×630.

**Tt Pah2 overexpression and purification**. *Tt* Pah2 plasmids were transformed into BL21 (DE3) RIPL cells (Agilent Technologies, Cat. No. 230280) for protein overexpression. Cells were grown in Terrific Broth to an OD$_{600}$ of 1.5 and then incubated with 100 μM isopropyl β-D-1-thiogalactopyranoside (IPTG) at 15 °C overnight before harvesting. Cell pellets were lysed in buffer A comprised of 50 mM Tris pH 8, 500 mM NaCl, 60 mM imidazole, 5% glycerol, and 2 mM beta-mercaptoethanol (βME) and lysates centrifuged at 16,000 RPM. *Tt* Pah2 protein was purified using a HisTrap FF column and eluted in buffer B, with an increased imidazole concentration of 300 mM. Due to saturation, this purification step was repeated three times using the column flow through to maximize yield. If using the pET28 vector, the N-terminal His-tag was not cleaved and immediately followed by size exclusion chromatography. If using the ppSUMO vector, the N-terminal His-SUMO fragment was cleaved using ULP-1 protease overnight at 4 °C. Proteins were applied to a Superdex 75 26/60 HiLoad column (GE Healthcare) equilibrated with 20 mM Tris pH 8, 150 mM NaCl, 5% glycerol, 10 mM βME, and 1 mM dithiothreitol (DTT). Purified protein was concentrated to 7–10 mg/mL, flash-frozen, and stored at −80 °C. All mutants were expressed in ppSUMO and purified using the same method (Supplementary Fig. 4).

**Se-Met Tt Pah2**. Selenomethionine (Se-Met) *Tt* Pah2 was generated by transforming the *Tt* Pah2/pET28 plasmid into B834 (DE3) cells (Novagen, Cat. No. 69-041-3), which are auxotrophic for methionine. An overnight culture grown in Luria Broth was added to M9 minimal media supplemented with 19 standard

amino acids, L-selenomethionine (50 mg/L), and Kao and Michayluk Vitamin Solution 100× (Sigma-Aldrich). Cells were grown to $OD_{600}$ of 0.6 and induced with 100 µM IPTG at 15 °C overnight before harvesting. Se-Met Tt Pah2 was purified as described for the native protein.

**Mouse lipin-2 overexpression and purification.** The mouse lipin-2 N-Lip C-Lip fusion ppSUMO plasmid was transformed into BL21 (DE3) RIPL cells (Agilent Technologies, Cat. No. 230280) for protein overexpression. Cells were grown in Terrific Broth to an $OD_{600}$ of 1.5 and then incubated with 100 µM IPTG at 15 °C overnight before harvesting. Cell pellets were lysed in buffer A comprised of 50 mM Tris pH 7.5, 500 mM NaCl, 60 mM imidazole, 5% glycerol, and 2 mM βME and lysates centrifuged at 22,000 RPM. Mouse lipin-2 protein was purified using a HisTrap FF column and eluted in buffer B, with an increased imidazole concentration of 500 mM. The N-terminal His-SUMO fragment was cleaved using ULP-1 protease overnight at 4 °C. Cleaved protein was applied to a Superdex 75 26/60 HiLoad column (GE Healthcare) equilibrated with 20 mM Tris pH 7.5, 150 mM NaCl, 5% glycerol, 10 mM βME, and 1 mM DTT. Purified protein was concentrated to 1 mg/mL, flash-frozen, and stored at −80 °C.

**Crystallization and data collection.** Crystals of His-tagged native Tt Pah2 (10 mg/mL) were grown by hanging drop vapor diffusion using 0.2 M $Ca(NO_3)_2$, 15% PEG 8000, and 0.1 M MES pH 6 at room temperature. Crystals of untagged native Tt Pah2 (7 mg/mL) were grown by hanging drop vapor diffusion using 0.2 M $Mg(NO_3)_2$ and 10% PEG 3350. Micro-seeds using a seed-bead (Hampton research) were used to generate single crystals. Se-Met Tt Pah2 (7 mg/mL) crystals were grown in a reservoir solution of 0.2 M $Ca(NO_3)_2$ and 20% PEG 8000 with the addition of 1:50 diluted micro-seeds. Solutions containing 0.2 M $Ca(NO_3)_2$ or 0.2 M $Mg(NO_3)_2$, 18% PEG 8000, 0.1 M MES pH 6, and 30% glycerol were used as cryoprotectants. Native diffraction data were collected at Brookhaven National Lab NSLS II AMX beamline 17-ID-1 for untagged lipin grown in magnesium conditions, and FMX beamline 17-ID-2 for His-tagged lipin grown in calcium conditions. Se-Met SAD data were collected at the Advanced Photon Source GM CAT 23ID-B beamline at Argonne National Lab in 15° wedges. All data were processed using xia2[47] DIALS[48] in CCP4[49]. Both the calcium and magnesium data sets were highly anisotropic, which is reflected in the data collection and refinement statistics.

**Structure determination and refinement.** Phasing was carried out in Phenix[50] using Autosol[51] with 22 of the 24 Se-Met sites identified, and an initial model generated using Autobuild[52]. After manual modeling in coot[53] and several refinements in Phenix Refine[54], the nearly complete model was used as a search model in Phaser[55] for molecular replacement with the 3.0 Å native calcium data set. Additional model building in Coot and refinement in Phenix produced the final model (Table 1, PDB code: 6TZY). The final calcium model contained four Tt Pah2 molecules in the asymmetric unit. Electron density for three of the four Tt Pah2 molecules was well resolved with the fourth Tt Pah2 molecule (chain D) having poorly resolved electron density for portions of the HAD-like domain. The magnesium structure was phased by molecular replacement with the calcium structure, placing two molecules in the asymmetric unit. The final model was produced by manual model building in coot and refinement in Phenix (PDB code: 6TZZ). Metal ions were modeled based on the strongest positive peak in an $F_O-F_C$ difference map after initial refinement and geometry and distance restraints were used in refinement.

**PAP assay.** Mixed micelles containing 10 mol% nitrobenzoxadiazole-phosphatidic acid (NBD-PA) (Cat. No. 9000341, Avanti Polar Lipids) and Triton X-100 were generated in a buffer containing 100 mM TRIS pH 7.5, 10 mM $MgCl_2$, 1 mM βME, and 100 mM NaCl. Micelles were incubated with 1 µg Tt Pah2 or with 7.5 ng mouse lipin 2 for 1 h at 37 °C. All enzyme assays were conducted in technical duplicates and experimental triplicates. All reactions were linear with respect to time and protein concentration (Supplementary Fig. 5). Reactions were quenched with the addition of 300 µL of HPLC grade 1:1 $CHCl_3$/MeOH (Sigma-Aldrich), followed by vortexing, and centrifugation at 2000 rpm. The organic phase was removed, dried under $N_2(g)$, and resuspended with 100 µL mobile phase B. Chromatographic separation was achieved utilizing an Agilent 1100 Series HPLC. Conditions were optimized using a Peek Scientific C-8 column (3 µm particle, 3.0 × 150 mm). Mobile phase A consisted of HPLC grade water containing 0.2% formic acid (Fisher Chemical) and 1 mM ammonium formate (Sigma-Aldrich, mass spec grade). Mobile phase B consisted of HPLC grade methanol containing 0.2% formic acid and 1 mM ammonium formate.

**Liposome sedimentation assay.** Liposomes were made by five freeze-thaw cycles and vortexing. The liposomes were then sonicated until a homogeneous solution was achieved, generating small unilamellar liposomes. The liposomes were extruded through a 100 nm membrane (Avestin) for 15 passes to ensure size uniformity. Liposomes were composed of 20 mol% phosphatidic acid (POPA) or 20 mol% phosphatidylserine (POPS), 40 mol% phosphatidylethanolamine (POPE), and 40 or 60 mol% phosphatidylcholine (POPC) (Avanti Polar Lipids) at a final concentration of 1 mM in a buffer containing 100 mM NaCl, 50 mM Tris pH 7.5, and 1 mM βME. Fifty microliters of liposomes was mixed with 50 µL of 1 µg protein and incubated at room temperature for 20 min. Liposomes were sedimented at

55,000 rpm (Beckman TLA 100.3, 163,348 × g) for 1 h and supernatants were removed. The soluble fraction was removed, the pellet was resuspended in an equal volume of buffer, and both samples resolved on SDS-PAGE. All liposome sedimentation assays were conducted in experimental triplicates.

**HDX-MS.** HDX-MS reactions were performed in a similar manner as is outlined in previous publications[39,40]. In brief, HDX reactions were conducted in a final reaction volume of 20 µL with a Tt Pah2 concentration of 24 pmol. Prior to the addition of deuterated solvent, the lipid containing Tt Pah2 samples were allowed to incubate with either 4 µL of 4 mg/mL lipid vesicles (20% phosphatidic acid, 60% phosphatidylcholine, 20% phosphatidylethanolamine) or 4 µL of the corresponding lipid buffer: 20 mM HEPES pH 6.5, 100 mM KCl, 10 mM $MgCl_2$. After a 2-min incubation period, $D_2O$ buffer (100 mM NaCl, 20 mM Bis-Tris pH 6.5, 10 mM $MgCl_2$, 93% $D_2O$ (v/v)) was added and the reaction allowed to proceed for 3, 30, 300, 3000 or 10,000 s at 18 °C before being quenched with ice-cold acidic quench buffer, resulting in a final concentration of 0.6 M guanidine-HCl and 0.9% FA post quench. All conditions and timepoints were created and run in triplicate. Samples were flash-frozen and stored at −80 °C until injection onto an ultra-performance liquid chromatography (UPLC) system for proteolytic cleavage, peptide separation, and injection onto a QTOF for mass analysis.

Protein samples were rapidly thawed and injected onto an UPLC system kept in a cold box at 2 °C. The protein was run over two immobilized pepsin columns (Applied Biosystems; Porosyme 2-3131-00) and the peptides were collected onto a VanGuard Precolumn trap (Waters). The trap was eluted in line with an ACQUITY 1.7 µm particle, 100 × 1 mm² C18 UPLC column (Waters), using a gradient of 5–36% B (Buffer A 0.1% formic acid, Buffer B 100% acetonitrile) over 16 min. Mass spectrometry (MS) experiments were performed on an Impact QTOF (Bruker) and peptide identification was done by running tandem MS (MS/MS) experiments run in data-dependent acquisition mode. The resulting MS/MS data sets were analyzed using PEAKS7 (PEAKS) and a false discovery rate was set at 1% using a database of purified proteins and known contaminants. HDExaminer Software (Sierra Analytics) was used to automatically calculate the level of deuterium incorporation into each peptide. All peptides were manually inspected for correct charge state and presence of overlapping peptides. Deuteration levels were calculated using the centroid of the experimental isotope clusters. Differences in exchange for a peptide were considered significant if they met all three of the following criteria: >4% change in exchange, >0.3 Da difference in exchange, and a $p$ value < 0.01 using a two-tailed Student's $t$ test. All compared samples were set within the same experiment. The full deuterium incorporation data can be found in the source data, with all data analysis parameters as mandated by the IC-HDX-MS guidelines[56] shown in Supplementary Table 1. The mass spectrometry proteomics data have been deposited to the ProteomeXchange Consortium via the PRIDE[57] partner repository with the data set identifier PXD017575.

**Thermal shift assay.** Wild-type Tt Pah2 and mutants were diluted in 100 mM Tris, pH 7.5, 100 mM NaCl, 10 mM $MgCl_2$ to 1 µg with 2× SYPRO Orange (Sigma-Aldrich). SYPRO orange fluorescence was measured on an RT-PCR instrument with a temperature increase of 1 °C/min. Fluorescence was normalized to the highest measured value and then fit to a nonlinear Boltzmann sigmoidal curve. Melting temperature was determined from the associated temperature of 50% normalized fluorescence. Melting temperatures of mutants were compared to wild-type Tt Pah2 to assess their effect on stability and performed in technical duplicates and experimental triplicates.

**Reporting summary.** Further information on research design is available in the Nature Research Reporting Summary linked to this article.

## Data availability

Data supporting the findings of this manuscript are available from the corresponding author upon reasonable request. A reporting summary for this Article is available as a Supplementary Information file. Coordinates and structure factors have been deposited in the Protein Data Bank under accession codes 6TZY and 6TZZ. The mass spectrometry proteomics data have been deposited to the ProteomeXchange Consortium via the PRIDE[57] partner repository with the data set identifier PXD017575. All other data are available from the authors on request. The source data underlying Figs. 1c, 4a–e, g, 5c–f, and 6c are provided as a Source Data file.

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

## Acknowledgements

We thank the staff at the AMX and FMX (NSLS-II) and GM-CAT (APS) beamlines for assistance during data collection, George Carman and Gil-Soo Han (Rutgers) for sharing yeast strains and plasmids, Georg Jander and Pavan Kumar (Cornell) for sharing *S. tuberosum* cDNA, and Vitaly Citovsky and Benoit Lacroix (Stony Brook) for sharing *N. tabacum* cDNA. Confocal laser-scanning microscopy was performed at the Advanced Light Microscopy/Spectroscopy Laboratory and the Leica Microsystems Center of Excellence at the California NanoSystems Institute at UCLA with funding support from NIH Shared Instrumentation Grant S10OD025017 and NSF Major Research Instrumentation grant CHE-0722519. This work was supported by the National Institutes of Health grants R35 GM128666 (M.V.A.), P01 HL090553 (K.R.), and P01 HL028481 (K.R.), the American Heart Association grants 17SDG33410860 (M.V.A.), 19PRE34450192 (V.I.K.), and 18POST34060200 (H.W.), the NSERC Discovery Grant NSERC-2014-05218 (J.E.B.), the Canadian Institutes of Health Research (J.E.B., New Investigator Award), the Michael Smith Foundation for Health Research (J.E.B., Scholar Award 17686), and the Chhabra-URECA award (J.A.B.).

## Author contributions

V.I.K. performed all protein purifications, crystallization experiments, liposome sedimentation assays, and enzyme assays. R.M.H. performed all HDX-MS experiments. R.M.H. and J.E.B. analyzed all HDX-MS data. H.W. performed all cellular localization studies. J.A.B. performed yeast spot assays and constructed key plasmids. V.I.K. and M.V.A. determined and refined the final crystal structures. V.I.K., K.R., J.E.B., and M.V.A. contributed intellectual and strategic input. K.R., J.E.B., and M.V.A. supervised work. V.I.K. and M.V.A. wrote the initial manuscript with contributions from R.M.H., H.W., J.E.B., and K.R. V.I.K., K.R., J.E.B., and M.V.A. edited the final manuscript. All authors approved the final manuscript.

## Competing interests

The authors declare no competing interests.
