## [Peer Review File · Nature Communications]

Reviewers' Comments:

Reviewer #1:

Remarks to the Author:

The manuscript by Khayyo et.al. presents a structural analysis of the *Tetrahymena thermophila* lipin/Pah phosphatidic acid phosphatase (Tt Pah2), a protein that catalyzes the formation of diacylglycerol through phosphatidic acid hydrolysis. The author first describes its crystal structure before employing hydrogen-deuterium exchange mass spectrometry (HDX-MS) and biochemical assays to characterize its association with membrane. The structure reveals that part of the C-Lip domain forms the catalytic site while its remaining part co-folds with the N-Lip region to form an immunoglobulin-like domain. HDX-MS and biochemical data suggested the presence of an N-terminal amphipathic helix regulating the protein association to PA-enriched membranes. Finally, the authors performed a structural mapping of disease mutations that are associated with catalytic dysfunction of Tt Pah2 for most of them.

Overall the manuscript is well-written and the methodology to obtain the structural data is robust. I did not find any flaws that should prohibit its publication.

I have minor points and comments:

Page 4: Could the author explain the rationale for using fatty acid treatment in the subcellular localization experiments?

Page 6: The authors mention that "this active site rearrangement induced by Calcium is consistent with the Magnesium dependence of lipin/pah PAPs." The meaning of this sentence is not entirely clear to me. Could the authors be more precise to explain how the differences between Ca and Mg structures could impact the enzymatic activity?

Was a "short" humanized version of Tt Pah2 tested in enzymatic assay? It could be a better system to analyze catalytic dysfunction of disease mutations.

Reviewer #2:

Remarks to the Author:

In this study the authors have crystallographically solved the structure of Th Pah2 showing that N-Lip co-folds with C-Lip forming a previously unknown Ig-like domain. This provides new insights into the role of N-lip in lipin/Pah function. Moreover, the authors used HDX-MS to identify that an amphipathic helix is essential for membrane binding. Overall, the manuscript is clearly presented and advances knowledge of a currently elusive system at molecular level. However there are a few issues that the authors may want to address:

a) The HDX-MS experiments are of high quality. The authors have been very transparent with their data and there is no doubt that these experiments complement very well the structural data. While the presented HDX-MS in the presence and absence of liposome containing PA is a clever experiment, I was wondering if as a control the authors could also carry out the same experiment in the deleted version? It might be that no association will be identified as the amphipathic helix has been deleted, but it may be worth exploring other potential interactions or the absence of them.

b) Following the above point, I believe, what it will also be particularly exciting if the authors can use HDX-MS to map the conformational changes of the disease-mutations. I however acknowledge that this might be a lot of work and possibly beyond the scope of this work. Nevertheless, this may lead to important information regarding the conformational mechanisms of diseases.

c) In Fig. 4 (HDX-MS), it may improve the readability if the authors label peptides found with significant deprotection to their respective regions/domains.

d) While the paper is overall well-written, I found figure 2 and the related text a bit confusing. For instance, for someone who is not an expert in this particular system, it is quite difficult with all the terminology and possibly simplifying this might benefit the paper. In particular, in such figure it will be useful if the authors show where is the catalytic domain which they refer to in the relative text.

Reviewer #3:

Remarks to the Author:

This study describes a structure/function analysis of a phosphatidic acid (PA) phosphate belonging to the lipin family of proteins, which hydrolyzes PA to DAG subsequently used for TG synthesis and other functions. These proteins are important because of their mutation and inactivation in inflammatory and metabolic diseases. Here, the crystal structure of a homologue from *Tetrahymena thermophila* (TtPah2) is determined to 2.66 Å. TtPah2 shares significant seq. id to the human enzyme (36-42%) for the catalytically important regions, but lacks a long linker between the conserved N-lip and C-lip domains, making it more amenable for structural studies. The structure reveals as expected, a two-domain configuration, with an N-terminal IgG-like domain and C-terminal HAD domain packed together. The HAD domain is similar to those previously determined as is the disposition of catalytic residues. The novel aspect of the structure comes from the IgG-like domain that is unexpectedly composed of peptide segments from both the N- and C-lip domains, implying that in the human and mouse orthologues, the long intervening linkers would emanate from within the IgG-like scaffold. The authors used HDX to identify surface regions of the enzyme important for membrane interaction and identify a potentially membrane-induced amphipathic helix at the N-terminus. Mapping of disease mutations provides some insight into the molecular underpinnings for loss of lipin function in human diseases.

This study is well done, and the paper is pleasingly concise, well written and the figures are clear and informative. The work provides important structural insights into the architecture of lipin PAP enzymes and their role in disease. There are no major concerns with the work, but the following points should be addressed by the authors.

1. Did the authors try to express and measure activity from lipins with linkers (eg. human, mouse, yeast) with the linker regions deleted analogously to the structure determined here? This would support the notion that higher lipins function with the same domain architecture as revealed here.
2. Related to the above, what is the nature of the interface between N-lip and C-lip? Would this interface be conserved in the human enzyme?
3. Is the amphipathic helix thought to only tether the enzyme to membrane or do the authors expect the hydrophilic face to pack against the protein?
4. It was not made clear how the enzyme processes PA molecules. Are the PA fatty acid tails recognized by the enzyme and is the enzyme surface compatible for this? Or is only the phosphate portion recognized? If only the phosphate group is recognized, how specific is the enzyme for PA? Can it act on other lipid phosphates such as sphingosine-1-phosphate, ceramide-1-phosphate? Some details regarding substrate binding should be discussed.
5. How was the identity of the metals in the various structures determined? Are the observed geometry and distances consistent with calcium and magnesium ion binding?
6. The B-factor (~ 74) for the 2.66 Å structure is high. Generally, at this resolution one would expect a Wilson B-factor on the order of ~ 50 . Notable, the lower 3Å resolution Mg structure has a lower Wilson B. Perhaps the data cut-off for the Ca structure was too lax or the data was highly anisotropic? This is also reflected in the R-free of the Ca structure being ranked in the lower

percentile range for similar structures. Additionally, the overall Rmerge's (i.e. presumably Rsym) of ~17% and ~20% seem high, particularly given that it is synchrotron data.

7. A few more details regarding the structure solution might be included in the methods, eg. the number of SeMet incorporated into the protein, how many sites were found by AutoSol, etc. Was TLS used in the refinement?

8. In the abstract, it won't be clear to the general reader, what is meant by 'capless' until after the Results section. Consider something like "...catalytic domain that lacks additional capping domains for productive...".

Reviewer #1 (Remarks to the Author):

The manuscript by Khayyo et al presents a structural analysis of the *Tetrahymena thermophila* lipin/Pah phosphatidic acid phosphatase (Tt Pah2), a protein that catalyzes the formation of diacylglycerol through phosphatidic acid hydrolysis. The author first describes its crystal structure before employing hydrogen-deuterium exchange mass spectrometry (HDX-MS) and biochemical assays to characterize its association with membrane. The structure reveals that part of the C-Lip domain forms the catalytic site while its remaining part co-folds with the N-Lip region to form an immunoglobulin-like domain. HDX-MS and biochemical data suggested the presence of an N-terminal amphipathic helix regulating the protein association to PA-enriched membranes. Finally, the authors performed a structural mapping of disease mutations that are associated with catalytic dysfunction of Tt Pah2 for most of them.

Overall the manuscript is well written and the methodology to obtain the structural data is robust. I did not find any flaws that should prohibit its publication. I have minor points and comments:

We thank the reviewer for their thoughtful reading of the manuscript. The suggestions that were given greatly improve the paper. We have provided our responses in blue font and have highlighted changes in the manuscript by using track changes and shown in purple font.

Page 4: Could the author explain the rationale for using fatty acid treatment in the subcellular localization experiments?

Fatty acid treatment has been shown to dephosphorylate mammalian lipins and promote ER association. To clarify our rationale in the manuscript, we have updated the text as follows (new text is underlined):

“In HEK293 cells, *Tt* Pah2 localized exclusively to the endoplasmic reticulum (ER) independent of fatty acid treatment, which is known to cause translocation of mammalian lipins to the ER^{31,32} (Fig. S3).”

Page 6: The author's mention that "this active site rearrangement induced by Calcium is consistent with the Magnesium dependence of lipin/pah PAPs." The meaning of this sentence is not entirely clear to me. Could the authors be more precise to explain how the differences between Ca and Mg structures could impact the enzymatic activity?

Thank you for bringing this to our attention. We have now clarified our original statement to read:

“In the structure determined in the presence of calcium, a Ca²⁺ ion was modeled in the active site in an altered position in comparison to the Mg²⁺ ion, with Asp272 coordinating the Ca²⁺ ion, along with the sidechains of Asp146 and Asn268 (Fig. 3b). The altered location of the Ca²⁺ ion and subsequent rearrangement of key active site

residues could prevent proper binding and hydrolysis of PA, which is consistent with the magnesium dependence of lipin/Pah PAPs.”

Was a "short" humanized version of Tt Pah2 tested in enzymatic assay? It could be a better system to analyze catalytic dysfunction of disease mutations.

Thank you for the suggestion. This suggestion was also similar to a question from Reviewer #3, and we have addressed both comments together by cloning, overexpressing, purifying, and testing the activity of a new construct of mouse lipin 2 that resembles the domain architecture of Tt Pah2 with the N-Lip and C-Lip domains directly fused. As can now be seen in a new Figure 6, the N/C-Lip truncation of mouse lipin 2 is catalytically active and at a higher activity than Tt Pah2. Importantly, we believe this verifies that the functional unit of lipin/Pah PAPs is the N-Lip/C-Lip architecture of Tt Pah2 established by our study.

While we believe this new lipin 2 construct would be an improvement on Tt Pah2 to biochemically analyze the disease-associated mutations, we believe this still represents a model system and the best system would be the full-length mammalian lipins. However, purifying these would require extensive effort and in our opinion is outside the scope of this manuscript. Nevertheless, we believe our analysis of Tt Pah2, which we have structurally defined, provides important insight into the human mutations. The updated text is as follows:

“The N-Lip and C-Lip represent the functional catalytic unit of lipin/Pah PAPs. To support the notion that mammalian lipin PAPs function with the same domain architecture observed in *Tt* Pah2, we generated analogous constructs of mouse lipin 1 and mouse lipin 2 that directly fused the N-Lip and C-Lip regions (**Fig. 6a**). We focused on characterizing the mouse lipin 2 construct because it was more easily purified from *E. coli* (**Fig. S6**). The mouse lipin 2 N-Lip C-Lip fusion protein retained PAP activity *in vitro* and was an order of magnitude more active than *Tt* Pah2 (**Fig. 6b**). Notably, a similar construct of *Sc* Pah1 that fuses the N-Lip and C-Lip regions has been shown to retain PAP activity *in vitro*, as well as maintain cellular function if a Trp-containing motif that is C-terminal to the C-Lip is retained⁴³. Together, this suggests the functional catalytic unit of lipin/Pah PAPs is represented by the domain architecture observed in *Tt* Pah2 with the N-Lip and C-Lip domains co-folding to form the Ig-like and HAD-like domains. In support of this, the major structural interfaces between the N-Lip and C-Lip regions are largely conserved between *Tt* Pah2, mammalian lipins, and *Sc* Pah1 (**Fig. 6c**) as observed by near complete sequence conservation of the core regions of the Ig-like fold, such as the β A, β A', β B, and β B' strands of the N-Lip (**Fig. S2b**).”

Reviewer #2 (Remarks to the Author):

In this study the authors have crystallographically solved the structure of Tt Pah2 showing that N-Lip co-folds with C-Lip forming a previously unknown Ig-like domain. This provides new insights into the role of N-lip in lipin/Pah function. Moreover, the authors used HDX-MS to identify that an amphipathic helix is essential for membrane binding. Overall, the manuscript is clearly presented and advances knowledge of a currently elusive system at molecular level. However there are a few issues that the authors may want to address:

We thank the reviewer for the kind comments, and bringing these important points to our attention. We have edited the manuscript and figures to help clarify the issues that were raised. We have provided our responses in blue font and have highlighted changes in the manuscript by using track changes and shown in purple font.

a) The HDX-MS experiments are of high quality. The authors have been very transparent with their data and there is no doubt that these experiments complement very well the structural data. While the presented HDX-MS in the presence and absence of liposome containing PA is a clever experiment, I was wondering if as a control the authors could also carry out the same experiment in the deleted version? It might be that not association will be identified as the amphipathic helix has been deleted, but it may be worth exploring other potential interactions or the absence of them.

Thank you. We have considered performing this experiment with the Δ Helix construct as a control but have decided that it is unlikely to provide new information. As shown in **Fig. 4b**, removing the amphipathic helix completely abolishes liposome interaction. Thus, generating and using a Δ Helix-D146A construct to perform HDX-MS is expected to not reveal any protected peptides, as the Δ helix construct does not interact with liposomes.

b) Following the above point, I believe, what it will also be particularly exciting if the authors can use HDX-MS to map the conformational changes of the disease-mutations. I however acknowledge that this might be a lot of work and possibly beyond the scope of this work. Nevertheless, this may lead to important information regarding the conformational mechanisms of diseases.

Thank you, we find this suggestion very interesting to pursue. However, we believe the best system to carry out these experiments would be with the full-length human or mouse lipins, with the corresponding disease-associated mutations, as this would directly measure the associated conformational changes. While we are currently expanding our studies to the full-length human and mouse lipins, this work is still in progress and represents a significant amount of work, which we believe would be outside the scope of the current manuscript.

c) In Fig. 4 (HDX-MS), it may improve the readability if the authors label peptides found with significant deprotection to their respective regions/domains.

Thank you for noting this, we have now labeled the peptides with their enzyme domains as suggested, and this is now shown in a modified **Fig. 4c**.

d) While the paper is overall well-written, I found figure 2 and the related text a bit confusing. For instance, for someone who is not an expert in this particular system, it is quite difficult with all the terminology and possibly simplifying this might benefit the paper. In particular, in such figure it will be useful if the authors show where is the catalytic domain which they refer to in the relative text.

We agree with your assessment and thank you for ensuring our manuscript is accessible to a general audience. We have now modified the text and figures to achieve more clarity and clearly explain the nomenclature associated with the cap domains commonly found in HAD-like catalytic domains and how this applied to the structure of the Tt Pah2 HAD-like catalytic domain, its association with membranes, and the potential modes of substrate recognition with its membrane-embedded substrate PA. Figure 2 has been updated, and the modified text/paragraph now reads:

“Catalytic mechanism. Key active site residues to hydrolyze PA to DAG are clustered at the putative membrane interface in the typical topological positions of HAD members (**Fig. 3a**). In the structure determined in the presence of magnesium, a Mg^{2+} ion was modeled in the typical position for HAD phosphatases with correct geometry and distances expected for Mg^{2+} coordination. This Mg^{2+} ion is coordinated by the sidechain of the first Asp residue of the DxDxT motif (Asp146), as well as by the sidechain of Asn268 and the mainchain carbonyl oxygen of Asp148. As expected in the absence of the substrate PA, the sidechain of Asp148 (the second Asp residue of the DxDxT motif) is directed away from the Mg^{2+} ion and forms a salt bridge with Arg193. Other residues predicted to stabilize the trigonal bipyramidal transition state^{20,34} are present in the expected positions, with Lys244 directed towards the Mg^{2+} ion and Ser191 adjacent (**Fig. 3a**).

The structure suggests that lipin/Pah PAPs utilize a similar catalytic mechanism as proposed for other HAD phosphatases^{20,34} with Asp146 serving as a nucleophile to attack PA and form the trigonal bipyramidal transition state, the surrounding residues stabilizing the transition state through electrostatic interactions, and Asp148 activating a water molecule for nucleophilic attack to liberate DAG and inorganic phosphate (**Fig. 3c**).

In the structure determined in the presence of calcium, a Ca^{2+} ion was modeled in the active site in an altered position in comparison to the Mg^{2+} ion, with Asp272 coordinating the Ca^{2+} ion, along with the sidechains of Asp146 and Asn268 (**Fig. 3b**). The altered location of the Ca^{2+} ion and subsequent rearrangement of key active site residues could prevent proper binding and hydrolysis of PA, which is consistent with the magnesium dependence of lipin/Pah PAPs.”

Reviewer #3 (Remarks to the Author):

This study describes a structure/function analysis of a phosphatidic acid (PA) phosphate belonging to the lipin family of proteins, which hydrolyzes PA to DAG subsequently used for TG synthesis and other functions. These proteins are important because of their mutation and inactivation in inflammatory and metabolic diseases. Here, the crystal structure of a homologue from *Tetrahymena thermophila* (TtPah2) is determined to 2.66 Å. TtPah2 shares significant seq. id to the human enzyme (36-42%) for the catalytically important regions, but lacks a long linker between the conserved N-lip and C-lip domains, making it more amenable for structural studies. The structure reveals as expected, a two-domain configuration, with an N-terminal IgG-like domain and C-terminal HAD domain packed together. The HAD domain is similar to those previously determined as is the disposition of catalytic residues. The novel aspect of the structure comes from the IgG-like domain that is unexpectedly composed of peptide segments from both the N- and C-lip domains, implying that in the human and mouse orthologues, the long intervening linkers would emanate from within the IgG-like scaffold. The authors used HDX to identify surface regions of the enzyme important for membrane interaction and identify a potentially membrane-induced amphipathic helix at the N-terminus. Mapping of disease mutations provides some insight into the molecular underpinnings for loss of lipin function in human diseases.

This study is well done, and the paper is pleasingly concise, well written and the figures are clear and informative. The work provides important structural insights into the architecture of lipin PAP enzymes and their role in disease. There are no major concerns with the work, but the following points should be addressed by the authors.

We thank the reviewer for carefully reading our manuscript and providing extremely helpful suggestions that improved our paper. We have provided our responses in blue font and have highlighted changes in the manuscript by using track changes.

1. Did the authors try to express and measure activity from lipins with linkers (eg. human, mouse, yeast) with the linker regions deleted analogously to the structure determined here? This would support the notion that higher lipins function with the same domain architecture as revealed here.

Thank you for this suggestion. We have performed our NBD-PA activity assay using a truncated version of mouse lipin 2, which only contains the N-Lip and C-Lip regions. The enzyme is catalytically active without the ~500 amino acid linker region that separates the two domains. Interestingly, the activity is higher than Tt Pah2. We have added this enzyme assay to a new Figure 6. We have addressed this in our manuscript as follows:

“The N-Lip and C-Lip represent the functional catalytic unit of lipin/Pah PAPs. To support the notion that mammalian lipin PAPs function with the same domain architecture observed in *Tt* Pah2, we generated analogous constructs of mouse lipin 1 and mouse lipin 2 that directly fused the N-Lip and C-Lip regions (Fig. 6a). We focused on characterizing the mouse lipin 2 construct because it was more easily purified from

E. coli (Fig. S6). The mouse lipin 2 N-Lip C-Lip fusion protein retained PAP activity *in vitro* and was an order of magnitude more active than *Tt* Pah2 (Fig. 6b). Notably, a similar construct of *Sc* Pah1 that fuses the N-Lip and C-Lip regions has been shown to retain PAP activity *in vitro*, as well as maintain cellular function if a Trp-containing motif that is C-terminal to the C-Lip is retained⁴³. Together, this suggests the functional catalytic unit of lipin/Pah PAPs is represented by the domain architecture observed in *Tt* Pah2 with the N-Lip and C-Lip domains co-folding to form the Ig-like and HAD-like domains. In support of this, the major structural interfaces between the N-Lip and C-Lip regions are largely conserved between *Tt* Pah2, mammalian lipins, and *Sc* Pah1 (Fig. 6c) as observed by near complete sequence conservation of the core regions of the Ig-like fold, such as the β A, β A', β B, and β B' strands of the N-Lip (Fig. S2b)."

2. Related to the above, what is the nature of the interface between N-lip and C-lip? Would this interface be conserved in the human enzyme?

The interface is largely conserved with the core of the Ig-like domain that is formed by interactions with the N-Lip and C-Lip having a high degree of sequence identity and homology. We incorporated these thoughts into the new paragraph in the point above and now have a panel in a **new Figure 6** that shows the positions of the residues in *Tt* Pah2 that share homology (either sequence identity or sequence similarity) with human/mouse lipins and *Sc* Pah1.

3. Is the amphipathic helix thought to only tether the enzyme to membrane or do the authors expect the hydrophilic face to pack against the protein?

This is an interesting point. The packing of the helix cannot be definitively determined since we do not have structural information from crystallography for the N-terminus of *Tt* Pah2. However, we believe its main role is membrane tethering, and it perhaps may pack against the protein to bring it closer to the membrane interface. We thank the reviewer for noting this and we have added this to the manuscript as follows:

"While the hydrophobic face of the amphipathic helix of *Tt* Pah2 clearly interacts with the membrane, it is not clear if the hydrophilic face of this helix packs against the protein or remains exposed to solvent."

4. It was not made clear how the enzyme processes PA molecules. Are the PA fatty acid tails recognized by the enzyme and is the enzyme surface compatible for this? Or is only the phosphate portion recognized? If only the phosphate group is recognized, how specific is the enzyme for PA? Can it act on other lipid phosphates such as sphingosine-1-phosphate, ceramide-1-phosphate? Some details regarding substrate binding should be discussed.

We thank the reviewer for bringing these points to our attention. We have not performed this experiment with *Tt* Pah2 to say definitively that our construct has specificity for only PA. However, PA specificity has been determined for human lipins. We now cite the appropriate reference in the modified paragraph below (Donkor et al, JBC 2007).

The tails of the PA substrate are hidden in the membrane, and the active site of *Tt* Pah2 is very shallow. As lipin/Pah PAPs can catalyze a variety of PA molecules with different chain lengths, this is consistent with the structure, where we strongly suspect that PA recognition is driven by interaction with the glycerol-backbone of PA, and not through interaction with the fatty acid tails.

We now discuss this in the results section when we introduce the catalytic domain, along with a more detailed description of the architecture/nomenclature of HAD phosphatase (based on a question from Reviewer 2). The new paragraphs now read:

“Catalytic mechanism. Key active site residues to hydrolyze PA to DAG are clustered at the putative membrane interface in the typical topological positions of HAD members (**Fig. 3a**). In the structure determined in the presence of magnesium, a Mg^{2+} ion was modeled in the typical position for HAD phosphatases with correct geometry and distances expected for Mg^{2+} coordination. This Mg^{2+} ion is coordinated by the sidechain of the first Asp residue of the DxDxT motif (Asp146), as well as by the sidechain of Asn268 and the mainchain carbonyl oxygen of Asp148. As expected in the absence of the substrate PA, the sidechain of Asp148 (the second Asp residue of the DxDxT motif) is directed away from the Mg^{2+} ion and forms a salt bridge with Arg193. Other residues predicted to stabilize the trigonal bipyramidal transition state^{20,34} are present in the expected positions, with Lys244 directed towards the Mg^{2+} ion and Ser191 adjacent (**Fig. 3a**).

The structure suggests that lipin/Pah PAPs utilize a similar catalytic mechanism as proposed for other HAD phosphatases^{20,34} with Asp146 serving as a nucleophile to attack PA and form the trigonal bipyramidal transition state, the surrounding residues stabilizing the transition state through electrostatic interactions, and Asp148 activating a water molecule for nucleophilic attack to liberate DAG and inorganic phosphate (**Fig. 3c**).

In the structure determined in the presence of calcium, a Ca^{2+} ion was modeled in the active site in an altered position in comparison to the Mg^{2+} ion, with Asp272 coordinating the Ca^{2+} ion, along with the sidechains of Asp146 and Asn268 (**Fig. 3b**). The altered location of the Ca^{2+} ion and subsequent rearrangement of key active site residues could prevent proper binding and hydrolysis of PA, which is consistent with the magnesium dependence of lipin/Pah PAPs.”

5. How was the identity of the metals in the various structures determined? Are the observed geometry and distances consistent with calcium and magnesium ion binding?

Thank you for bringing this to our attention. In both structures, density was observed in the active site that indicated presence of a metal ion and the metals were manually modeled into the density based on the strongest peak in the difference map. In the structure determined in the presence of magnesium, a Mg^{2+} ion was modeled in this

position with the expected geometry and distances for magnesium binding. We have now modified the text to state this:

“In the structure determined in the presence of magnesium, a Mg^{2+} ion was modeled in the typical position for HAD phosphatases with correct geometry and distances expected for Mg^{2+} coordination.”

For the structure determined in the presence of calcium, the electron density with the strongest peak in the difference map suggested the calcium ion to be near Asp272, which was not involved in Mg^{2+} coordination. Geometry and distance restraints were used in refinement. The text has been modified accordingly:

Results section: “In the structure determined in the presence of calcium, a Ca^{2+} ion was modeled in the active site in an altered position in comparison to the Mg^{2+} ion, with Asp272 coordinating the Ca^{2+} ion, along with the sidechains of Asp146 and Asn268 (Fig. 3b).”

Methods section: “Metal ions were modeled based on the strongest positive peak in an F_o-F_c difference map after initial refinement and geometry and distance restraints were used in refinement.”

6. The B-factor (~74) for the 2.66 Å structure is high. Generally, at this resolution one would expect a Wilson B-factor on the order of ~50. Notable, the lower 3Å resolution Mg structure has a lower Wilson B. Perhaps the data cut-off for the Ca structure was too lax or the data was highly anisotropic? This is also reflected in the R-free of the Ca structure being ranked in the lower percentile range for similar structures. Additionally, the overall Rmerge's (i.e. presumably R_{sym}) of ~17% and ~20% seem high, particularly given that it is synchrotron data.

You are correct. Both the calcium and magnesium data sets indeed suffered from significant anisotropy, which is reflected in the data collection and refinement statistics. We agree that we extended the resolution cutoff for the calcium structure too far, and have now cut the resolution at 3.0Å. We are in the process of updating the pdb entry with the lower resolution mtz used for refinement and will do so before release, We will also still include the complete high resolution data in the unmerged mtz file, for any to access.

We have also now included a sentence in the methods section that explicitly states the data was anisotropic:

Methods section: “Both the calcium and magnesium data sets were highly anisotropic, which is reflected in the data collection and refinement statistics.”

7. A few more details regarding the structure solution might be included in the methods, eg. the number of SeMet incorporated into the protein, how many sites were found by AutoSol, etc. Was TLS used in the refinement?

Thank you, we have updated the paper to explicitly state the number of sites found. TLS refinement was not used.

“Phasing was carried out in Phenix⁴⁸ using Autosol⁴⁹ with 22 of the 24 Se-Met sites identified,”

8. In the abstract, it won't be clear to the general reader, what is meant by 'capless' until after the Results section. Consider something like "...catalytic domain that lacks additional capping domains for productive...".

Thank you, this is a good suggestion. We actually decided to remove this particular phrase from the abstract to make room for the new data about mouse lipin-2. We also broke that particular sentence into two sentences because we felt it was a run on sentence and didn't communicate the major structural findings to a general audience. The new abstract reads as follows:

“Lipin/Pah phosphatidic acid phosphatases (PAPs) generate diacylglycerol to regulate triglyceride synthesis, lipoprotein assembly, and cellular signaling. Inactivating mutations cause rhabdomyolysis, autoinflammatory disease, and aberrant fat storage. Disease-mutations cluster within the conserved N-Lip and C-Lip regions that are separated by 500-residues in humans. To understand how the N-Lip and C-Lip combine for PAP function, we determined crystal structures of *Tetrahymena thermophila* Pah2 that directly fuses the N-Lip and C-Lip. The structure reveals the N-Lip combines with the C-Lip to form an immunoglobulin-like domain. The remainder of the C-Lip forms the catalytic domain from the haloalkanoic acid dehalogenase-like family. An N-Lip C-Lip fusion of mouse lipin-2 is catalytically active, which suggests mammalian lipins function with the same domain architecture as *Tt* Pah2. HDX-MS identifies an N-terminal amphipathic helix essential for membrane association. Disease-mutations directly disrupt catalysis or destabilize the protein fold. This study sheds light on the mechanisms of lipin/Pah PAP function, membrane association, and lipin-related pathologies.”

Reviewers' Comments:

Reviewer #1:

Remarks to the Author:

The authors have addressed my points.

Reviewer #2:

Remarks to the Author:

Happy for this manuscript to be accepted

Reviewer #3:

Remarks to the Author:

All of my comments have been satisfactorily addressed.